# Evaluation of the Interfacial Interaction Ability between Basalt Fibers and the Asphalt Mastic

**DOI:** 10.3390/ma15228209

**Published:** 2022-11-18

**Authors:** Bangwei Wu, Zhaohui Pei, Peng Xiao, Keke Lou

**Affiliations:** 1College of Civil Science and Engineering, Yangzhou University, Yangzhou 225100, China; 2Research Center for Basalt Fiber Composite Construction Materials, Yangzhou 225127, China

**Keywords:** fiber asphalt mastic, interfacial interaction ability, evaluation indicator, pull-out test, surface energy

## Abstract

The interfacial properties between the asphalt mastic and fibers plays an essential role in the fiber-enhanced asphalt mixture properties. However, there is a lack of comprehensive studies on the indicators to evaluate the interfacial interaction ability of fibers with the asphalt mastic. Therefore, this paper selected three types of basalt fibers (denoted as A-BF, B-BF and C-BF) coated with different impregnating agents to prepare the fiber asphalt mastic. The Dynamic Shear Rheometer (DSR) test-based indicators, pull-out strength, and adhesion work were used to access the fiber asphalt mastic interfacial interaction ability. The differences between different indicators were compared and analyzed. The results show that all the selected indicators in this paper can effectively reflect the different fiber asphalt mastic interfacial properties. The evaluation results with different indicators are consistent. The interfacial interaction between fibers and the asphalt mastic increases with increasing temperature. The evaluation result with adhesion work is the most accurate. However, the pull-out strength test is simple, and the test result correlates well with adhesion work, which can be adopted daily to evaluate the fiber asphalt mastic interfacial properties.

## 1. Introduction

Using polymer modifiers and fibers is the most common method to enhance the engineering properties of asphalt mixtures [1], and a growing number of researchers have paid attention to the role of fibers in asphalt mixtures [2]. Back in the 1970s, some states in the U.S.A. used fibers to improve the crack resistance of asphalt mixtures [3]. Since the 1990s, lignin and mineral fibers have been used in large numbers in Stone Mastic Asphalt (SMA) to stabilize asphalt [4,5]. From then on, researchers have tested more types of fibers in asphalt mixtures, for example, nylon fiber [6], polyester fiber [7], carbon fiber [8], glass fiber [9], etc. In recent years, more and more researchers have used basalt fiber for asphalt mixtures due to its excellent engineering properties [10,11]. According to the American Transportation Research Board investigation [12], introducing fibers to asphalt mixtures has many benefits such as increasing the tensile strength, improving the fatigue resistance and rutting resistance, and reducing life cycle costs.

To design fiber-reinforced asphalt mixtures properly, many researchers have investigated the reinforcement mechanism of fibers. Shen et al. [13] evaluated the bonding ability of basalt fibers (BF) with different lengths to asphalt mastic using strip tensile tests and dynamic shear rheological tests. It was concluded that BF form a fiber network in the asphalt mastic, which can effectively relieve stress concentrations and thus retards cracking. This viewpoint has been endorsed by many other researchers [14,15]. Using the bending beam rheological (BBR) test and the dynamic shear rheological (DSR) test, Xing [16] study the effect of fibers with different physical forms on the rheological of the asphalt mastic. He concluded that the flocculent fiber could enhance the stability of the mastic by toughening the mastic. Similar phenomena have been observed in other studies [17,18]. As for the role of fibers in asphalt mixtures, the most common explanation is that fibers can form a complex fiber network by lapping each other, and this three-dimensional network can act as a bridge to transfer and dissipate stresses, thus improving the strength and toughness of the asphalt mixture [15,16,17,18].

In addition to factors such as fiber length and dosage, the fiber–asphalt interfacial interaction ability affects the properties of fiber–asphalt mixtures as well. Xiang conducted several studies in modifying fiber surface to improve asphalt mastic properties [19,20]. For example, he coated the fiber surface with a silane coupling agent. The DSR results and infrared spectrum test results of fiber asphalt mastics before and after modification indicated that the silane coupling agent enhanced the interfacial chemical bonding between fibers and asphalt, resulting in an overall increase in the asphalt mechanical performance [19]. Similar conclusion was also drawn by Liu [21] and Lou [22]. Yoo [23] also argued the importance of interface properties in fiber asphalt concrete. He found that the interfacial bonding property between fiber and asphalt affect the tensile strength significantly. Park proposed a hypothesis to explain the enhancement mechanism of fiber [24]. He believed that the reinforcing effect of fibers on the asphalt concrete is caused by two aspects—the fiber–aggregate interlocking and fiber–binder adhesion. However, most of the current research has overlooked the latter aspect. Although some researchers have noticed the importance of the former mechanism [25,26], the fiber–asphalt interfacial bonding has not been studied enough, especially the indicators for characterizing the fiber–asphalt interfacial bonding ability have not been compared and evaluated.

Therefore, this paper aims to assess the interfacial interaction capability of fiber and the asphalt mastic through a series of evaluation indicators, and to recommend a feasible test method to determine the fiber–asphalt bonding ability. In this paper, three short-cut BF coatings with different types of impregnating agents were selected. The DSR test, fiber pull-out test, and contact angle meter were used to determine the interfacial interaction in fiber asphalt mastic, and different evaluation indicators were compared and analyzed.

## 2. Materials

### 2.1. Asphalt Binder

Polymer modified asphalt (PG 64–22) is used in this study. Its technical performances are presented in Table 1.

### 2.2. Mineral Powder

This paper used limestone powder in the asphalt mastic. The technical performance of the mineral powder is presented in Table 2.

### 2.3. Fiber

This paper used three short-cut BFs for the test, named A-BF, B-BF, and C-BF. The length and diameter of the three BFs are 6 mm and 17 μm. They are almost the same except for the impregnating agents on their surfaces. Impregnating agent is a necessity coating the fibers to avoid a brittle break of fibers. Different impregnating agents have different chemical compositions and can cause their different adhesion properties with asphalt [27]. Table 3 shows the composition of the three basalt fiber impregnating agents.

## 3. Test Method

### 3.1. Preparation of Asphalt Mastic

To better reflect the quality ratio of engineering road fiber to asphalt binder in the actual mixture, the relevant research conclusions on fiber-reinforced asphalt mixture were adopted, and the fiber content is calculated to be 5% of the asphalt binder mass. The powder to asphalt ratio used in this study is 1.0. The following procedures were followed when preparing fiber asphalt mastic. Firstly, the mineral powder and fibers were dried at 120 °C for not less than 1.5 h. The asphalt binder was heated to 175 °C to simulate the mixing temperature in the actual project. Secondly, the asphalt was gradually mixed with the fiber and mineral powder combination in three sections. The fiber asphalt mastic was stirred at 1000 rpm for 30 min. To ensure uniform dispersion of BFs in asphalt mastics, the mineral filler and fibers were premixed before being added to the asphalt. Thirdly, the mastic was stirred for not less than 20 min at a speed of 500 rpm to eliminate the air bubbles inside it. Throughout the procedure, the asphalt mastic temperature was maintained at 175 °C ± 5 °C.

### 3.2. DSR Test

The AASHTO T315 standard describes the DSR test protocols that were used in this work. The dynamic shear complex modulus as well as the phase angle are used to express test results. The test piece has a 25 mm diameter and a 1 mm thickness. There were three parallel specimens in this research.

According to composite material theory, when an additive is introduced into the matrix, the interfacial interaction between the matrix and the additive will inevitably lead to changes in the rheological parameters of the matrix (e.g., phase angle, complex modulus, etc.), and then, the interfacial interaction ability between the matrix and the additive can be characterized according to the changes in the rheological parameters before and after the additive is introduced. There are many rheological parameters to reflect the interfacial interaction capability [28]. Based on DSR test results, several indicators were calculated according to Equations (1)–(4). These indicators are used to determine the interfacial interaction ability of asphalt mastics and the fiber. In the field of composite materials research, these indicators are used to characterize the interfacial interaction ability between matrix and additives [29,30,31]. In this paper, these indicators are cited to characterize the interfacial interaction ability of fiber and the asphalt mastic.
(1)$$A\left(\delta \right)=\frac{{\tan\delta }_{c}}{\left(1-\phi_{f}\right){\tan\delta }_{m}}-1$$
(2)$$B\left(\delta \right)=\frac{1}{\varphi_{f}}\times \left(\frac{{\tan\delta }_{m}}{{\tan\delta }_{c}}-1\right)$$
(3)$$A\left(G^{*}\right)=\frac{G_{c}^{*}-G_{m}^{*}}{\varphi_{f}\times \left(1.5G_{m}^{*}+G_{c}^{*}\right)}$$
(4)$$A\left(G^{\prime \prime }\right)=\frac{1}{\varphi_{f}}\times \left(1-\frac{G_{m}^{\prime \prime }}{G_{c}^{\prime \prime }}\right)$$
where:A(δ), B(δ), A(G*), A(G″) = indicators for assessing the interfacial contact capability of the fiber and the asphalt mastic,δ_c_ = phase angle of the fiber asphalt mastic,δ_m_ = phase angle of asphalt mastics without fibers,G_c_*, G_c_″ = the complex module and loss module of the fiber asphalt mastic,G_m_*, G_m_″ = the complex module and loss module of asphalt mastics without fibers,φ_f_ = volume fraction of fibers.

### 3.3. Pull-Out Test

This test is used to determine the bond strength between BF and the asphalt mastic. This test is carried out in a fiber–asphalt pull-out tester designed by the author’s research group, and the instrument has been patented. The test is performed according to the following steps. First, one end of a bundle of continuous fibers is fixed in the center of a rectangular metal test mold (L × W × H = 20 cm × 1 cm × 4 cm), and the other end is fixed on the tensile equipment (illustrated in Figure 1). Then, the asphalt mastic was poured into the mold. The mold was placed at 50 °C for not less than 3 h. Then, according to our previous research on this instrument [32], the fibers were pulled off the asphalt mastic at a rate of 10 mm/min by a tester. A sensor records the tensile force during the pulling process. The pull-out strength is calculated by Equations (5) and (6). The specific details of the fiber asphalt pull-out tester are presented in Figure 2. In this research, four parallel specimens were prepared.
(5)$$\tau =\frac{F_{\max}}{S}\times 1000$$
(6)S=2×(b+h)×L
where τ = pull-out strength of the fiber and the asphalt mastic/kPa,F_max_ = maximum tensile strength/N,S = contact area between fibers and the asphalt mastic, i.e., the shear area/mm^2^,b = the length of the fiber specimen/mm, determined by an optical microscope,h = the width of the fiber specimen/mm, determined by an optical microscope,L = the length of fiber in the asphalt mastic/mm.

**Figure 1 materials-15-08209-f001:**
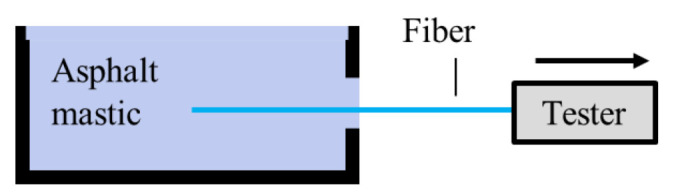
Schematic of pull-out test.

**Figure 2 materials-15-08209-f002:**
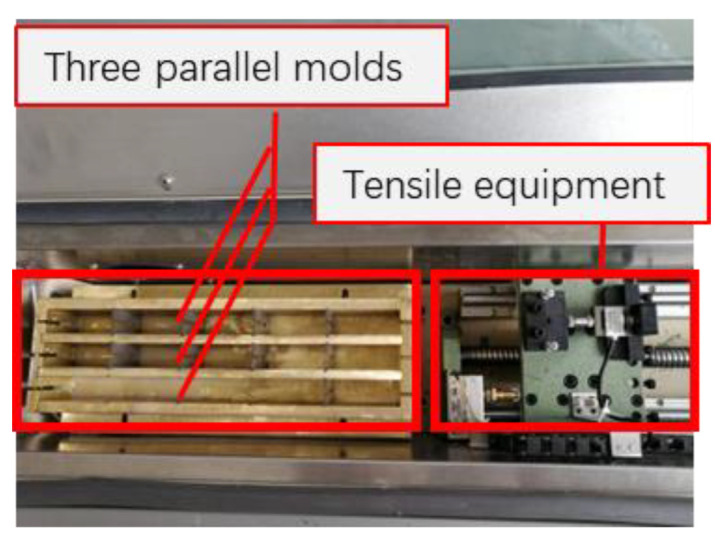
Internal structure of test machine (vertical view).

### 3.4. Contact Angle Test

A video optical contact angle measuring device (OCA 40, DataPhysics Instruments GmbH, Filderstadt, Germany) was used to perform this test, as seen in Figure 3. To obtain a flat surface of asphalt specimens used for testing, this paper adopted the following steps. The asphalt was heated to a uniform flow (approximately 170 °C); then, a 10 mm-thick rectangular glass plate was put into the melted asphalt vertically. The glass plate was then removed and suspended in an oven for 3 h, allowing the extra asphalt to drip down. The temperature in the oven was about 170 °C. The asphalt specimens are shown in Figure 4. Each specimen was tested in four different parts.

By referring to other researchers’ findings, the authors selected distilled water and glycol to measure the surface energy parameters of BF and the asphalt [33,34]. The polar and non-polar components of the surface energy of BF and the asphalt were determined by Equation (7) with the surface energy parameters. The adhesion work between asphalt binder and BF was further calculated by Equation (8) to evaluate the bonding ability between the two.
(7)$$1+\cos\theta =2\sqrt{\gamma_{s}^{d}}\left(\frac{\sqrt{\gamma_{l}^{d}}}{\gamma_{l}^{d}+\gamma_{l}^{p}}\right)+2\sqrt{\gamma_{s}^{p}}\left(\frac{\sqrt{\gamma_{l}^{p}}}{\gamma_{l}^{d}+\gamma_{l}^{p}}\right)$$
(8)$$W_{a}=2\sqrt{\gamma_{a}^{d}\gamma_{a}^{d}}+2\sqrt{\gamma_{f}^{p}\gamma_{f}^{p}}$$
where:$\gamma_{l}^{d}$, $\gamma_{l}^{p}$ = the non-polar and polar components of the surface energy of liquids, mJ/m^2^,$\gamma_{s}^{d}$,$\gamma_{s}^{p}$ = the non-polar and polar component of surface energy of solids, i.e., the BF or the asphalt mastic, mJ/m^2^,θ = solid-liquid surface contact angle, °,W_a_ = the adhesion work between the asphalt mastic and BF, mJ/m^2^,$\gamma_{a}^{d}$, $\gamma_{a}^{p}$ = the non-polar and polar component of the surface energy of the asphalt mastic, mJ/m^2^,$\gamma_{f}^{d}$,$\gamma_{f}^{p}$ = the non-polar and polar component of surface energy of BF, mJ/m^2^.

## 4. Results and Discussion

### 4.1. Indicators Based on DSR

In this study, the phase angle, loss modulus and complex modulus of the asphalt mastic with and without fibers were measured using temperature sweep tests. Using SPSS software (version 22.0), this paper carried out an analysis of variance (ANOVA) on the test results to determine whether there was a significant difference between the test results of the different fiber asphalt mastic. Then, the interfacial indicators were calculated by Equations (1)–(4). Figure 5 and Figure 6 present the results of the DSR test. Since the complex modulus and loss modulus exhibit the same rule, only the complex modulus test result is given here.

Figure 5 and Figure 6 show that after the addition of BF to the asphalt mastic, the complex modulus of the mastic increases while the phase angle decreases. The modulus of BF is larger than that of the asphalt mastic; the fibers can lap each other to form a network structure to prevent the asphalt mastic from flowing, resulting in the BF asphalt mastic higher modulus [15]. Additionally, the introduction of fibers makes the asphalt mastic more elastic, which reduces the asphalt mastic phase angle [17,18]. Furthermore, although having the same length, diameter, and dosage, the three BFs have different effects on the rheological characteristics of asphalt mastics. The order of effect is A-BF > C-BF > B-BF. Such difference is caused by the various impregnating agents on the fiber surface. The different impregnating agents caused different interfacial bond strengths between BF and the asphalt mastic and affected the rheological behavior of asphalt mastics [19].

Figure 7a–d show the interfacial indicators of BF asphalt mastics. These indicators were calculated based on the DSR results by Equations (1)–(4).

Smaller A(δ) and larger B(δ), A(G*), and A(G’’) indicate stronger interfacial bonding between the asphalt mastic and BF. The following points can be seen in Figure 7.
(1)When the four indicators were used to assess the interfacial interaction ability of BFs and the asphalt mastic separately, the same results were obtained, i.e., A-BF > C -BF > B-BF. As discussed previously, the three BFs were coated with various impregnating agents. When the basalt fibers are immersed into the asphalt mastic, the impregnating agents on the fiber surface adsorb the asphalt molecules to form interfacial layer gradually. In addition, chemical reaction between impregnating agent molecules and asphalt molecules may also occur [18]. Xiang [20] also noticed that the impregnating agent type influenced the fiber’s chemical bonding with asphalt, and it was due to the different physicochemical compositions of impregnating agents causing different reaction types and degrees with the asphalt mastic, which is manifested by the different interfacial interaction abilities between fibers and asphalt mastic.(2)A(δ) decreases with the temperature, B(δ), A(G*), and A(G’’) increase with the temperature, indicating that the interface interaction between basalt fiber and the asphalt mastic increases at higher temperatures. The higher the temperature, the greater the enhancement of the interfacial interaction ability. Liu [35] also found that the interfacial interaction ability increased with the temperature when using DSR-based indicators to evaluate the interfacial bond between the asphalt and the mineral powder. The higher the temperature, the more active the movement of the impregnating agent molecules and the asphalt molecules, resulting in an adequate reaction between the two [36]. At higher temperatures, the impregnating agent can wet the fiber surface more fully; on the other hand, the impregnating agent and asphalt mastic are more fully integrated, resulting in a stronger interfacial interaction between the fiber and the asphalt mastic. However, Liu [35] concluded that A(G*) is not sensitive to temperature and is not applicable to evaluating the mineral powder–asphalt interfacial interaction ability. From the data in Figure 7a–d, it can be observed that the four indicators evaluated in this paper are all sensitive to temperature, and all of them can effectively distinguish the different fiber–asphalt mastic interfacial interaction abilities.

### 4.2. Indicator Based on Pull-Out Test

The pull-out test was used in this study to evaluate the bond strength between BF and the asphalt mastic. The results are shown in Figure 8. Each test result is the average of four measurements. An ANOVA on the results was carried out to determine whether there was a significant difference between them in the first place.

Some scholars believe that the fiber asphalt mixes fail because the fibers are pulled out from the asphalt mastic [5]. Therefore, an excellent fiber–asphalt interfacial bonding state is the key to ensuring that the fiber plays a reinforcing role. The stronger the interfacial interaction ability of the fiber and the asphalt mastic, the higher the interfacial pull-out strength [6]. As seen in Figure 8, the pull-out strength of A-BF is the highest, followed by C-BF and B-BF, indicating that the interfacial interaction ability of A-BF with asphalt mastic is better than that of C-BF and B-BF, which is consistent with the evaluation results of DSR-based indicators. As previously discussed, when the impregnating agent on the fiber surface and the asphalt mastic contact each other, complex physicochemical reactions such as adsorption between them occur [18]. In this way, an interfacial layer is formed that bonds the fiber and asphalt together. Different impregnating agents will form interfacial layers with different bond strengths with the asphalt mastic. The data in Figure 8 shows that the interfacial pull-out strength between A-BF and asphalt mastic is 86.4% higher than that between B-BF and asphalt mastic, which proves that the type of impregnating agent has a crucial influence on the interfacial bond strength. Therefore, when selecting fibers for asphalt mixtures, besides considering the fiber type and fiber content, the evaluation of the fiber–asphalt interfacial interaction ability should also be considered.

### 4.3. Indicator Based on Surface Energy

According to thermodynamic theory, the interfacial interaction between two objects is largely related to surface energy; therefore, indicators based on surface energy can more accurately reflect the interfacial interaction ability of fibers and the asphalt mastic.

In this paper, the distilled water and glycol whose surface energy parameters are known were used to determine the surface energy parameters of BF and the asphalt mastic. The contact angles of BF and the asphalt mastic were determined, and then their surface energy parameters were calculated by Equation (7). The results are presented in Table 4. Further, the adhesion work between BF and the asphalt mastic was calculated by Equation (8), and the result is shown in Figure 9.

The following points can be seen from Table 4.
(1)The three basalt fibers exhibited different contact angles regardless of which test liquid was used. When using distilled water as the test liquid, the contact angles of A-BF, B-BF, and C-BF were 51.3°, 69.5°, and 54.6°, respectively. When ethylene glycol was used as the test fluid, the values were 25.4°, 56.3°, and 41.2° for the three fibers. The contact angle is the smallest for A-BF and the largest for B-BF. The contact angle characterizes the wetting ability of liquids on a solid surface. The smaller the contact angle, the easier the liquid wets and spreads on the solid surface [37]. According to the interfacial chemistry theory [38], the wetting and spreading of liquid on the solid surface is essential for forming the interfacial bonding layer, which implies that the liquid is more likely to form an interfacial bonding layer with A-BF. Therefore, the three basalt fibers’ different contact angles are one of the critical factors impacting their interfacial interaction ability with asphalt mastics.(2)The free surface energy of the three fibers ranked as A-BF > C-BF > B-BF. The material has the property of transforming from a high energy state to a low energy state. The larger the surface free energy of a material, the more unstable its surface state; material trends reduce the surface energy by reducing its surface area [39]. Therefore, the larger the surface free energy of a material, the stronger its adsorption ability. The different surface free energies of the three basalt fibers inevitably lead to their different abilities to adsorb asphalt molecules, which can be further illustrated by the calculated results of the adhesion work between the fibers and the asphalt mastic (seen in Figure 9).

Figure 9 shows the calculated results of the adhesion work between BF and asphalt mastics. The adhesion work between the three fibers and asphalt mastic is 36.46 mJ/m^2^, 28.89 mJ/m^2^ and 30.21 mJ/m^2^, i.e., A-BF > C-BF > B-BF. The larger the adhesion work between the two materials, the more work required to separate their interfacial layers, the stronger their interfacial interaction. From the results of the adhesion work, it can be seen that different impregnating agents can cause a difference of 1.26 times in the adhesion work between asphalt mastics and fibers. It indicates the importance of choosing a suitable impregnating agent for fibers and the necessity of assessing the interfacial interaction ability between the asphalt mastic and fibers.

### 4.4. Comparison of Different Indicators

The above analysis shows that all the evaluation indicators selected in this paper can effectively distinguish the different interfacial interaction abilities between the asphalt mastic and fibers. The adhesion work is a calculated value based on the thermodynamic theory, and many scholars believe that this index can accurately evaluate the interfacial bonding performance between two materials [37,38]. Other evaluation indicators, however, are more like a category of phenomenological indicators. Therefore, this paper uses the adhesion work as a benchmark value to analyze its correlation with other indicators. The results are shown in Figure 10.

The following points can be seen in Figure 10.
(1)The correlation coefficient between pull-out strength and adhesion work reached 0.90, indicating that the pull-out strength can accurately evaluate the interfacial interaction ability between fibers and asphalt mastic. Compared with testing the adhesion work between fiber and asphalt mastic, the pull-out test is simpler and cheaper, and can be used for the daily testing of the fiber–asphalt mastic interface interaction ability.(2)The correlation coefficients between the four DSR test-based evaluation indexes and the adhesion work differed greatly. The correlation coefficients between the modulus-based evaluation indexes (i.e., A(G*) and A(G’’)) and the adhesion work are larger than those between the phase angle-based evaluation indexes and the adhesion work (i.e., A(δ) and B(δ)), indicating that the modulus-based evaluation indexes are more accurate in evaluating the interfacial interaction capability within the fiber asphalt mastics. Some researchers [38] used DSR test-based indexes to assess the interfacial interaction capability of mineral powder and asphalt. They found that the evaluation indicators calculated with phase angle are more sensitive to the test variables (temperature, mineral powder content, et al.), so they suggested using the index based on phase angle to assess the mineral powder–asphalt interfacial interaction.However, in this paper, on the one hand, the evaluation indicators calculated with phase angle are more sensitive to the test variables. On the other hand, it can be challenging to ensure that fibers in the asphalt mastic are distributed uniformly. Therefore, the evaluation of phase angle-based index may possess more variability and exhibit a lower correlation coefficient with the adhesion work in the end.(3)The correlation coefficient between the DSR test-based indicators and the adhesion work was temperature-dependent. Taking A(G*) as an example, the correlation coefficients between A(G*) and adhesion work are 0.73, 0.75, 0.84, 0.97 and 0.82 at 64 °C, 70 °C, 76 °C, 82 °C, and 88 °C, respectively. Fiber asphalt mastic is a composite material. According to the composite material theory [39], the composite effect between the fiber and the asphalt mastic is related to many factors, such as the interfacial interaction between fiber and the asphalt mastic, the modulus ratio of fiber and the asphalt mastic, and so on. As a temperature-sensitive material, asphalt mastic must exhibit different modulus ratios with fibers at different temperatures and ultimately affect their composite effect. Therefore, A(G*) calculated from the complex modulus contains not only the effect of the interaction between the asphalt mastic and fibers, but also the effect of other factors, which causes the correlation between A(G*), and adhesion work is temperature-dependent.

## 5. Conclusions

In this paper, the interfacial interaction capability between the asphalt mastic and fibers is evaluated by several indicators. The conclusions can be drawn as follows:(1)After the addition of BF to the asphalt mastic, the complex modulus of the mastic increases while the phase angle decreases. Moreover, the impregnating agents critically impact the rheological properties of asphalt mastics.(2)All the four DSR test-based indicators can effectively reflect the different fiber–asphalt mastic interfacial interaction capability. Furthermore, the fiber–asphalt mastic interfacial interaction capability increases with the temperature raises.(3)Both pull-out strength and adhesion work can characterize the interfacial bonding performance of the fiber and the asphalt mastic. Both indicators can distinguish the different interfacial interaction capabilities between the asphalt mastic and fibers.(4)The pull-out strength test is simple and highly accurate compared with other indexes. It can be used daily to determine the interfacial interaction capability between the asphalt mastic and fibers.

The interfacial interaction between BF and asphalt mastic plays a key role in binder properties. This paper demonstrates that. However, the microscopic mechanisms of interfacial interaction are not yet well understood. In the future, more efforts should be tied to clarifying the interfacial mechanisms and proposing possible methods of improving the fiber–asphalt interfacial interaction.

## Figures and Tables

**Figure 3 materials-15-08209-f003:**
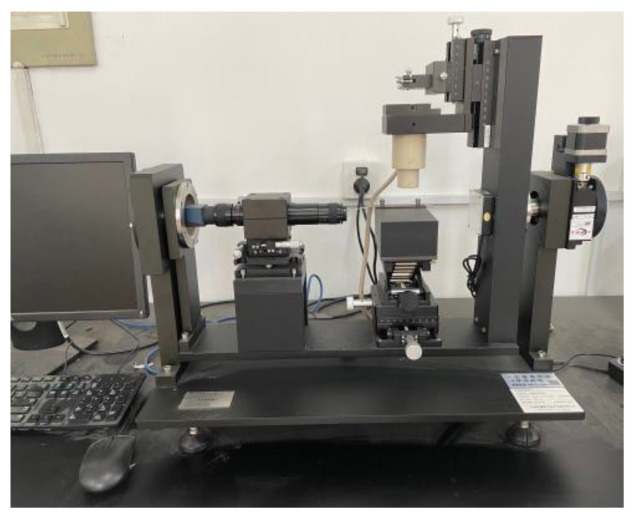
Video optical contact angle measuring instrument.

**Figure 4 materials-15-08209-f004:**
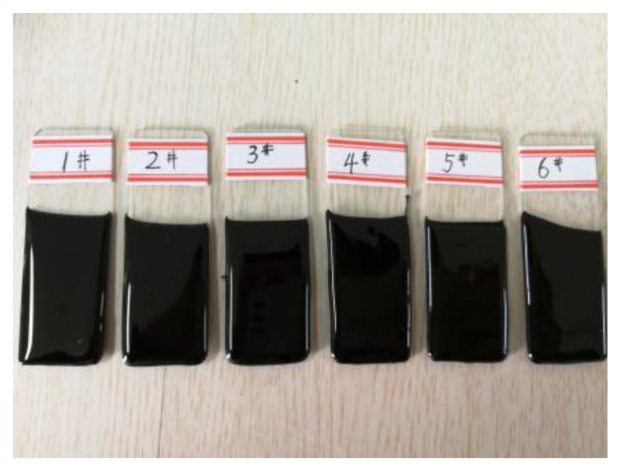
Asphalt specimen.

**Figure 5 materials-15-08209-f005:**
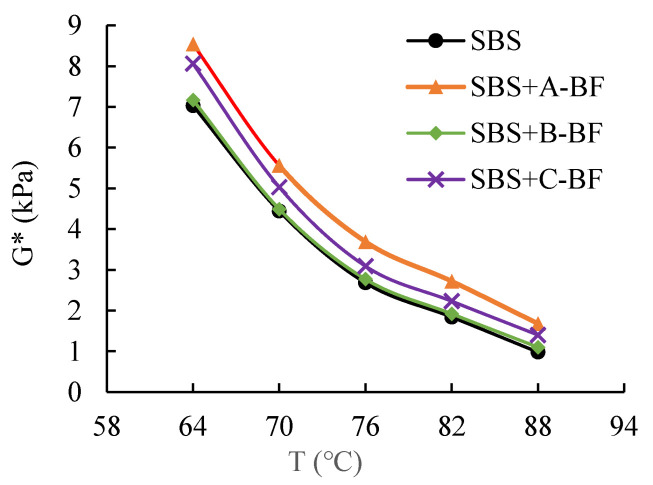
Results of the complex modulus.

**Figure 6 materials-15-08209-f006:**
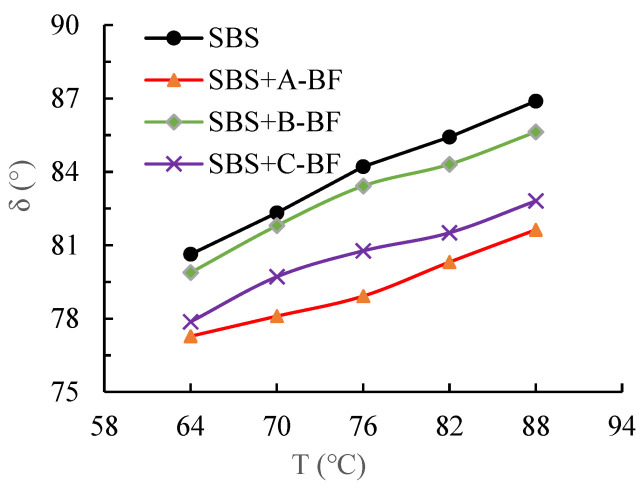
Results of the phase angle.

**Figure 7 materials-15-08209-f007:**
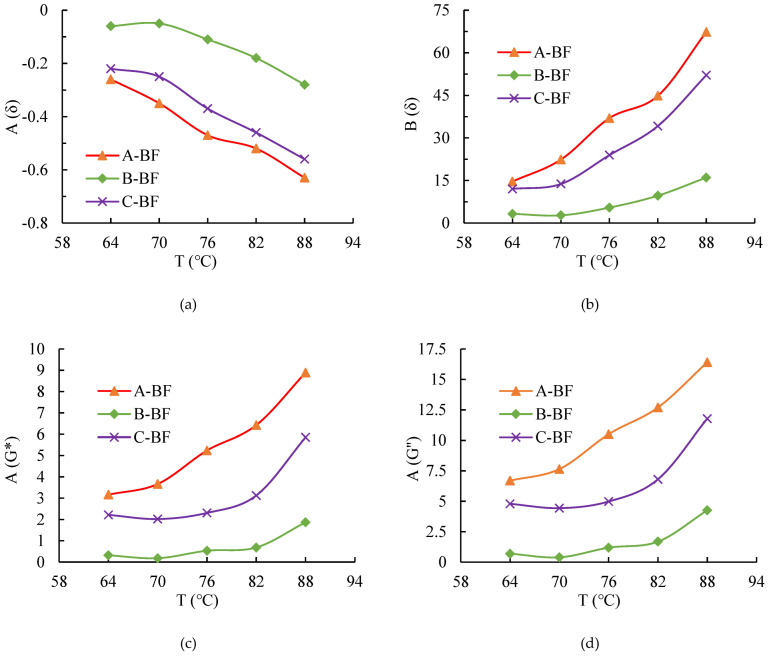
Indicators based on DSR: (**a**) A(δ), (**b**) B(δ), (**c**) A(G*) and (**d**) A(G’’).

**Figure 8 materials-15-08209-f008:**
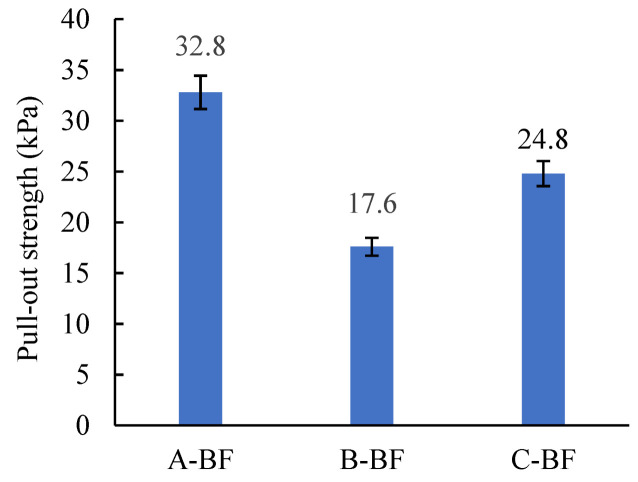
Pull-out strength of the basalt fiber asphalt mastic.

**Figure 9 materials-15-08209-f009:**
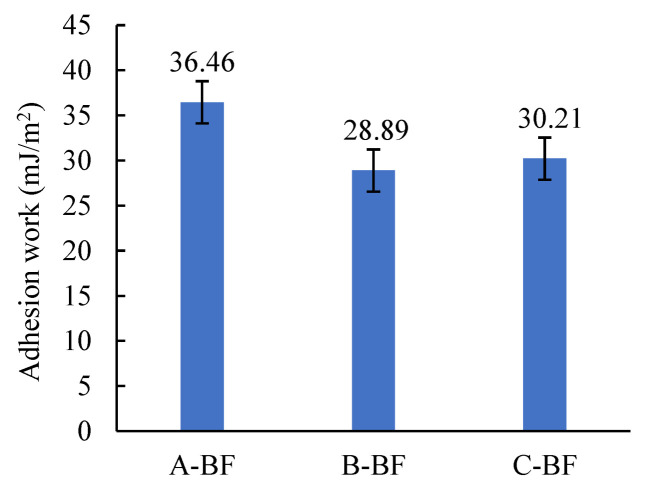
The adhesion work between BF and the asphalt mastic.

**Figure 10 materials-15-08209-f010:**
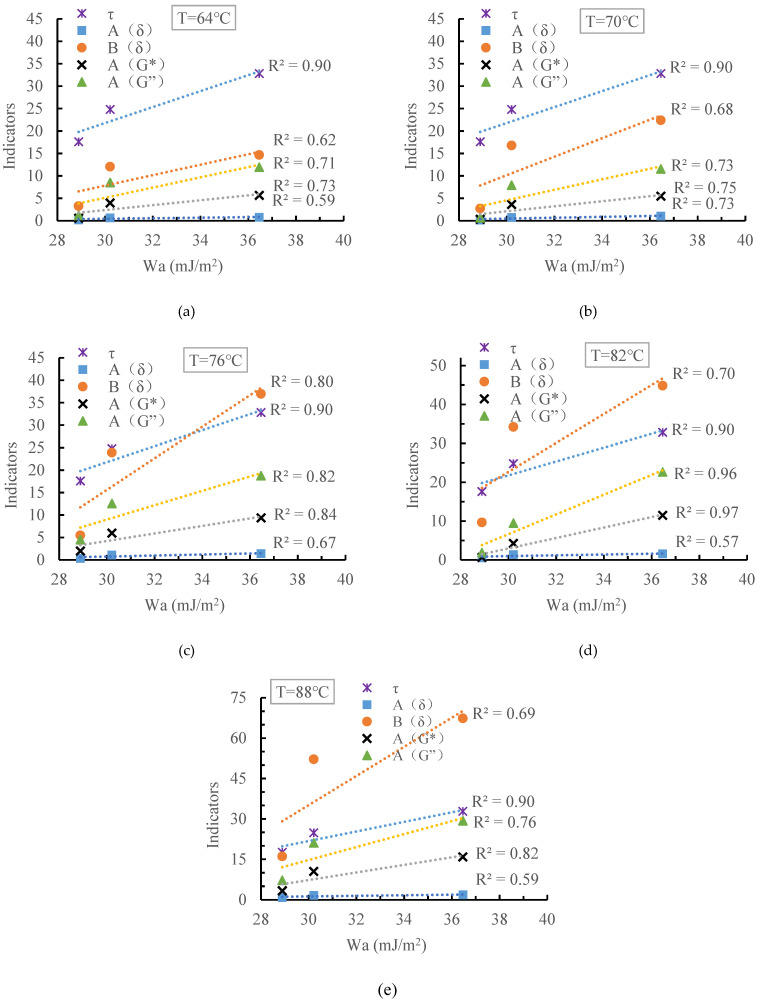
The correlation between adhesion work and other indicators at (**a**) 64 °C, (**b**) 70 °C, (**c**) 76 °C, (**d**) 82 °C and (**e**) 88 °C.

**Table 1 materials-15-08209-t001:** Technical performance of the asphalt binder.

Test Items	Test Results	Specification Requirements
Penetration (25 °C)/0.1 mm	71	60–80
Ductility (5 cm/min, 5 °C)/cm	48	>30
Softening point/°C	64	>55
Elastic recovery (25 °C)/%	76	>65
Kinematic viscosity (135 °C)/Pa.s	1.8	<3
Flash Point/°C	329	>230

**Table 2 materials-15-08209-t002:** Technical performance of the mineral powder.

Test Items		Test Results	Specification Requirements
Depending on the density/g/cm^3^		2.714	≥2.50
Water content/%		0.38	≤1.0
Plasticity Index		2.2	<4
Particle size range/%	<0.6	100	100
	<0.15	98.5	90–100
	<0.075	85.2	75–100

**Table 3 materials-15-08209-t003:** Composition of three kinds of basalt fiber impregnating agents.

Composition	Type-A	Type-B	Type-C
Main impregnating agents	polyvinyl acetate emulsion	polyester emulsion	Polyalcohol ester emulsion
Auxiliary impregnating agents	Water soluble epoxy resin	Epoxy emulsion	Water soluble epoxy resin
Coupling agents	KH550	A151	KH560
Lubricating agents	Polyoxyethylene Stearates	Polyoxyethylene Stearates	Ester

**Table 4 materials-15-08209-t004:** Surface energy of BF and the asphalt mastic (25 °C).

Types of Fiber	Contact Angle (°)		Surface Energy of Components (mJ/m^2^)		
	Distilled Water	Glycol	Surface Energy	Nonpolar Component	Polar Component
A-BF	51.3	25.4	47.95	14.32	33.63
B-BF	69.5	56.3	33.14	8.43	24.71
C-BF	54.6	41.2	46.26	7.73	38.53
Asphalt mastic	112.9	93.6	12.83	12.06	0.77

## Data Availability

The data presented in this study are available on request from the corresponding author.

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
