# Peer review of "Evaluation of the Interfacial Interaction Ability between Basalt Fibers and the Asphalt Mastic"

_materials, 2022, doi:10.3390/ma15228209_

Round 1

Reviewer 1 Report

This is a well-prepared manuscript examining the interaction of fibre-reinforced bituminous mastics by utilising three different approaches: DSR, pull-out test and contact angles by means of adhesion energy. Three types of fibres are used and coated with different agents which affect differently the results which are consistent between the three different techniques. The authors may want to consider some comments below before publication which might enhance even more the already high quality of this manuscript.

Line 12: What the parameters mean here is not clear (especially the functions A and B), so it would be recommended not to mention them in an abstract.

Line 15: It is not an appropriate wording that of ‘respectively’ since all of them are used to unravel the interfacial bonding. Please amend accordingly.

Line 20: It is not completely clear to what exactly the adhesion work is more accurate.

Line 28: Please replace the phrase ‘more and more’  to keep the wording on a more scientific level.

Line 44: There is a grammar mistake in the phrase ‘using and bonding’ of this sentence. Please take care of it.

Line 62: Please add the word ‘that’ after ‘believed’.

Line 69: Please add the word ‘a’ before ‘feasible’.

Line 70: Please replace ‘coating’ with ‘coatings’.

Line 76: What is the polymer content of this polymer-modified bitumen?

Table 1: Please replace the not smaller or not greater with just > and < respectively.

Line 85: Do all three fibres are 6 and 17 μm? Then which one is 6 and which is 17? From Figure 1 I suspect that types B and C are 17 and type A 6 μm. Please clarify in your revision.

Section 3.1. : Please provide a descriptive protocol rather than an instruction-like procedure for the potential reader.

Equations 1-4. : Please explain conceptually the background in a paragraph of these equations besides the given references to ensure a smooth follow-up for the reader.

Equation 5: How precise and how the contact area S is measured? Please add this info.

Figure 4: Please provide a better-quality image.

Lines 144-145: Please consult recent studies utilising also other meaningful rheological parameters related to distress phenomena of asphalt see i.e. https://doi.org/10.1617/s11527-022-01986-w

Line 154: Please discuss the effect of the different impregnating agents on the fibres’ surface and give more information about them for a coherent understanding of the differences they generate in the obtained results.

Line 166-167: Please provide a description/hypotheses of these complex physicochemical reaction mechanisms such as adsorption. 

Reviewer 2 Report

I recommend specifying in the title of the article the type of fiber used in the study (basalt fibers).

I recommend trying to answer the following questions in the article:

§  The asphalt binder (PG 64-22) used had an SBS-type polymer. How was the asphalt binder modified with SBS and how could this polymer affect the results? Why didn't the authors use a conventional asphalt binder (without polymer)?

§  What were the impregnating agents used? What are their (physical-chemical) properties and how did they affect the results and conclusions?

§  Why does the fiber-asphalt mastic interfacial interaction capability increase with the temperature raises? (Discuss in more depth).

§  The separate effect of BF and impregnating agent on the results is not clear.

§  Why didn't the authors evaluate the influence of aging on the performance and the interfacial interaction of mastics?

§  Why is the fiber dosage 5% of the asphalt binder mass?

§  175°C is a high mixing temperature. Why did the authors choose this temperature?

§  It is not clear Why the evaluation result with adhesion work is the most accurate?

I think the authors should improve the following paragraph: “The research only focused on extensive laboratory tests on the crack resistance of basalt fiber modified asphalt mixtures with different diameters. In future research, multiple characterization methods can be used for comprehensive analysis about of the relationship between the fiber diameter and the cracking mechanism of asphalt mixture”. The foregoing taking into account that the tests carried out were not extensive, the resistance to cracking was not the main property measured and further recommendations for future studies are needed.

Reviewer 3 Report

1-There are a lot of typos, grammatical and topographical errors throughout the text.

2 Authors have made statements throughout the paper without supporting by references, please carefully read the manuscript and add the proper references. 

3- Literature review part is weak. More research papers should be included in the literature review and in the Introduction. There are a large number of pioneer works regarding this subjects. 

4-Authors should explain total number of test specimens with their grouping and number of duplication for each test clearly in the revised paper.

5-There is no statistical analysis to clarify if the effect of each fiber type and size is meaningful. Please address this issue in the revised manuscript.

6 Mention the practical implementations and research limitations. 

7-why did you  select 10 mm/min loading rate ? Please discuss on the effect of loading rate on the pull out test. 

8- major conclusions should be explained rather than simply presenting the results

Round 2

Reviewer 1 Report

Thank you for the revise version. Good luck with this publication.

Reviewer 2 Report

The authors took into account my observations and made the respective changes

Reviewer 3 Report

The authors have satisfactorily responded to all my questions